# OpenReview forum: "RedDebate: Safer Responses Through Multi-Agent Red Teaming Debates"
_ICML.cc/2026/Conference — ICML 2026 regular_

### Official Review · Reviewer_v5yj · 2026-02-22

**Soundness:** 2
**Presentation:** 3
**Significance:** 3
**Originality:** 3
**Overall Recommendation:** 4
**Confidence:** 4

**Summary:**

This work created a framework named RedDebate. This framework aims to evaluate the effect of multi-agent debate on LLM safety. The idea is that given a scenario, the LLM can debate on the response and thus impact each other's answers. This response can be saved to a memory (in the prompt, finetuned via LoRA, used as a guardrail), thus impacting future generations with the aim of mitigating the mistakes in the future. They show that debating can reduce harmful rates and the importance of a memory mechanism. They show that some models sometimes move from safe to unsafe in the debate. They also investigated different debating strategies, including adding Devil agent and Angel agent, and an agent that asks critical questions.

**Compliance With Llm Reviewing Policy:**

Affirmed.

**Final Justification:**

This work investigates an interesting problem of the impact of debating on removing unsafe patterns.
I want to thank the authors again for the rebuttal. I will keep the score 4.

A concern from the author's last comments is that the preference for each setup is more complex and may also differ, given the model. While PReD improves upon self-revision, it seems small for some models (small ones), and as the authors state, it is large for others (big ones). Highlighting the need for further evaluation.
In addition, the memory mechanism, which, from the results, is an important component, lacks some understanding, such as its usefulness for more and less similar questions, and when it is needed.

**Key Questions For Authors:**

1. The authors write “often responses generated by debating agents can overlap or represent similar reasoning” in line 188. How different are the models in which cases they prefer to generate safe answer too?

2. The authors claim that "While self-revision leads to modest improvements as the number of revision rounds increases, the gains are generally limited. In contrast, PReD substantially reduces both total and per-agent error rates" (line 304).
 from sp -> sr, the total ER drop is 5.8 for HarmBench and 0.4 for CoSafe. And the additional drop from sr -> PReD is 4.1 for HarmBench and 0.5 for CoSafe.  What quantify as a modest or large improvement?

3. From figure 4, it seems that both debate and self-critique have a decrease in error rate. From my understanding (line 365 -”Debaters with memory steadily reduce error rates over successive round“) the debate setup also has a memory component. Was the effect mostly due to the debating or to the memory component?

**Limitations:**

yes

**Strengths And Weaknesses:**

Soundness: This paper investigates the impact of debating on removing unsafe patterns. It shows the importance of memory and of self-revision or debating on removing unsafe patterns. Ablation evaluation about the difference between self-reflection vs. debating can help understand the importance of debating.  It can also be beneficial to include an evaluation of how many iterations the memory lasts and its effect on more and less similar questions in the memory.

Presentation: The paper is presented well; it has a related work and uses known techniques in the framework.


Significance: The paper shows a framework to investigate multi-agent debating to decrease mistakes,  which is an important problem. Its result can impact future work in deciding how to best use the multi agent framwork to decrease mistakes.

Originality: This work creates a novel framework by integrating known debating strategies and memory methods. It shows how the different memory and debating strategies impact the mitigation of agent mistakes.

---

> ### Author Rebuttal · Authors · 2026-03-31
>
> **Comment 1:** \
> How much the debating models differ in the cases where they choose to generate safe responses, given the statement that agents’ answers can overlap or reflect similar reasoning.
>
> **Response:** \
> We thank the reviewer for the comment. In our setting, the debate is open-ended rather than multiple-choice, so there are often several distinct responses that would all be considered safe. For this reason, we do not impose a preference ordering among safe answers: if two agents produce different but equally safe responses, we treat them as equally valid.
>
> Our main objective is instead to identify and learn from cases where the debate exposes unsafe or weakly justified behavior. This is why overlap among safe responses is not problematic for our method. What matters is whether the interaction surfaces a vulnerability—i.e., whether an apparently safe trajectory can be pushed toward an unsafe one.
>
> To make this more informative, we introduce contrasting roles (e.g., devil/angel) to encourage diverse reasoning paths and stress-test the robustness of safe answers. In this process, we flag responses whose safety is brittle or insufficiently supported, while retaining responses that remain consistently safe under challenge.
>
> ---
>
> **Comment 2:** \
> The reviewer asks whether the improvement from self-revision (SR) to PReD is large enough to justify the use of the terms “modest” and “substantial” in the comparison.
>
> **Response:** \
> Thank you for the constructive comment. We will make the expressions more precise.
>
> Our intended comparison was with the standard-prompting baseline (SP), rather than with self-revision (SR). Conceptually, both SR and PReD are mechanisms that allow agents to recognize and correct an initially flawed response, so the relevant comparisons are:
>
> * **SP → SR**: self-correction through sequential self-revision
> * **SP → PReD**: correction through multi-agent deliberation
>
> Under this comparison, PReD yields a clearly larger reduction in error than SR. For instance, on HarmBench, total ER drops by 5.8 from **SP → SR**, compared to 9.9 from **SP → PReD**, a substantially larger improvement, roughly doubling the safety gain. On CoSafe, the overall reductions are smaller, but PReD still outperforms SR relative to SP (0.9 vs. 0.4), again representing about a twofold increase in safety gain.
>
> We agree that our original phrasing may have suggested the comparison was SR → PReD, which was not our intent. We will clarify this explicitly and replace the current wording with a more precise statement such as:
>
> > “Relative to the standard-prompting baseline, self-revision yields modest improvements, whereas multi-agent deliberation produces substantially larger error reductions—roughly doubling the safety gains.”
>
> This should remove the ambiguity and better align the claim with the reported numbers.
>
> ---
>
> **Comment 3:** \
> The reviewer asks whether the error reduction in Fig. 4 is driven by the debate mechanism or by the memory/guidance component.
>
> **Response:** \
> We thank the reviewer for the comment. As noted in the second column of line 312, the SR setup is inherently an extension of self-revision, where revisions are guided by a set of constitutions. These constitutions function as a static form of textual long-term memory. Consequently, in both Figure 4 and Figure 5, we intend for the mechanisms to receive external guidance: in the case of self-critique, this guidance comes from constitutions, while in the debate setting, it comes from textual feedback derived from agents’ previous mistakes.
>
> Therefore, the observed reduction in error rate should not be attributed solely to the presence of guidance or memory. Rather, our goal is to highlight the effect of the debate mechanism itself. We will revise the caption of Figure 4 to clarify this point and avoid potential confusion.

---

> > ### Author Rebuttal · Reviewer_v5yj · 2026-04-01
> >
> > Thank you for the rebuttal. I appreciate your answers. I will keep the score 4.
> >
> > A concern I still have, while it was somewhat answered, is regarding the importance of debate compared to SR or SC. I am still not convinced that the SR improvement is modest compared to the PReD. As the difference between them is a few percentage points. In addition, for a better understanding of the memory component, and when it is effective.

---

> > > ### Author Response · Authors · 2026-04-05
> > >
> > > We thank the reviewer for the thoughtful follow-up. Below, we address the individual comments.
> > >
> > > **Comment 1:**
> > >
> > > The improvement of PReD over SR.
> > >
> > > **Response:**
> > > Thank you for raising this point which helps clarify the purpose of comparing SR and PReD.
> > >
> > > We start by directly comparing the **error rate (ER)** of PReD and SR using the results in Table 1. To make this comparison clearer, we report the difference in error rate for each model and dataset below, where negative values indicate that **PReD improves over SR**:
> > >
> > > 1. Mistral/Llama/Phi models (Harmbench/CoSafe)
> > >
> > > |Method|Tot.|Mis.|Lla.|Phi|Tot.|Mis.|Lla.|Phi|
> > > |------|----|----|----|----|----|----|----|----|
> > > |SR|32.9|48.5|19.1|31.0|7.0|8.3|6.8|5.9|
> > > |PReD|28.8|37.2|21.3|27.9|6.5|7.5|6.3|5.7|
> > > |**Δ**|**−4.1**|**−11.3**|**+2.2**|**−3.1**|**−0.5**|**−0.8**|**−0.5**|**−0.2**|
> > >
> > > 2. Gemini/Qwen/R1 models (Harmbench/CoSafe)
> > >
> > > |Method|Tot.|Gem.|Qwe.|R1|Tot.|Gem.|Qwe.|R1|
> > > |------|----|----|----|----|----|----|----|----|
> > > |SR|37.0|29.1|38.8|43.2|7.9|9.2|6.4|8.0|
> > > |PReD|34.7|29.0|35.1|40.1|6.8|9.8|5.6|4.8|
> > > |**Δ**|**−2.3**|**−0.1**|**−3.7**|**−3.1**|**−1.1**|**+0.6**|**−0.8**|**−3.2**|
> > >
> > >
> > > Across the 16 model/dataset settings shown in Table 1, **PReD achieves lower error rate than SR in 14 out of 16 cases (87.5%)**. This shows that the improvement is not isolated to a small number of settings, but is instead broadly consistent. At the same time, we agree with the reviewer that the magnitude of the improvement is not always large. In some cases the gain is modest (for example, **CoSafe Phi: -0.2**, **Harmbench Gemini: -0.1**), and in a small number of cases SR performs better than PReD (for example, **Harmbench Llama: +2.2**, **CoSafe Gemini: +0.6**).
> > >
> > > Our point is not that **PReD dominates self-revision by a large margin in every setting**, but that **PReD consistently improves upon self-revision in most cases**, which already establishes the value of debate as a standalone mechanism. This is important because it shows that the benefit does not depend entirely on the extra guidance introduced in **DAReD** and **SReD**, and that debate is effective even in its direct form, while the more structured variants further strengthen this effect.
> > >
> > > We also think these results suggest an important model-dependent pattern. For smaller or weaker models, it may be harder to benefit from the dynamic and open-ended interaction required by debate. In such cases, the model may need to be pushed more explicitly toward correctness, which is exactly the motivation for the more structured debate variants we propose. This is consistent with the fact that **DAReD and SReD show larger and more consistent improvements over SR than PReD**.
> > >
> > > To support this interpretation **we extend the frontier-model results in Appendix C.2 to include SR results as well**. In this case, the difference between SR and PReD is much more substantial. Comparing total ER:
> > >
> > > - **Grok-4:** PReD = **2.5**, SR = 15.6
> > > - **R1-685B:** PReD = **12.6**, SR = 21.3
> > > - **Gemini-2.5-Pro:** PReD = **3.8**, SR = 6.0
> > >
> > > These results suggest that for **frontier models**, which are more capable of handling dynamic multi-agent interaction, the benefit of even open-ended debate can be much larger than for smaller models.
> > > We thank the reviewer for highlighting this point. In the revision, we will clarify this claim more carefully: **for smaller models, the more guided debate mechanisms such as DAReD and SReD appear better suited and more reliable, while for stronger frontier models, even the more open-ended PReD can lead to clear improvements over self-revision.**
> > >
> > > ---
> > >
> > > **Comment 2:**
> > > Role of the memory component and the conditions under which it is most effective.
> > >
> > > **Response:**
> > > The memory mechanism is a core component of our framework as it enables learning beyond a single debate instance. Specifically, it stores information from prior failures and outcomes, allowing the framework to improve not only during the current debate but also across future, related situations. In the sense, the memory component is most effective when the goal is to support cumulative learning: i.e., when similar errors or patterns recur across debates and past experience can inform future decisions. Without memory, the framework can only adapt within an individual debate; with memory, it can retain and reuse lessons from previous failures, which is essential for sustained improvement over time. **Memory allows the framework to avoid rediscovering the same corrections repeatedly, improving both efficiency and performance over time**. By contrast, without memory, the system can only adapt within the current debate and cannot accumulate experience across runs.
> > >
> > > We will clarify this role and its effectiveness regime in the paper.
> > >
> > > ---
> > >
> > > We sincerely thank the reviewer for the thoughtful feedback and engagement, which helps enhance the paper. If you think your main comments have been addressed, we humbly ask if you could further consider reflecting it in your score.

---

### Official Review · Reviewer_f48w · 2026-03-06

**Soundness:** 3
**Presentation:** 4
**Significance:** 3
**Originality:** 4
**Overall Recommendation:** 4
**Confidence:** 4

**Summary:**

This paper introduces a method named RedDebate, a multi-agent debate framework that lets LLMs collaboratively red-team their own responses to adversarial prompt and identify unsafe reasoning patterns. It explores three debate strategies and four long-term memory variants, with experiments showing up to 23.5 % unsafe response reduction.

**Compliance With Llm Reviewing Policy:**

Affirmed.

**Key Questions For Authors:**

Besides the weaknesses part above, have the authors considered the framework be extended to multimodal or tool-using agents?

**Limitations:**

The authors are suggested to provide cases such as retrieval failures in TLTM or catastrophic forgetting in CLTM.

**Strengths And Weaknesses:**

Strengths:

1.It has fair unification of multi-agent debate and automated red-teaming

2.It has clear ablation of debate strategies and memory types with strong quantitative gains

3.Its practical inference-time

4.It has separated evaluator from debaters to avoid shared delusion

Weaknesses:

1.The O(NT) inference cost of multi-round, multi-role debates on lightweight LLMs with limited context windows is not quantified, nor is any latency or throughput ablation presented

2.There is not enough exploration about the generalization of long-term memory or out-of-distribution robustness. The authors are suggested to provide cases such as retrieval failures in TLTM or catastrophic forgetting in CLTM

---

> ### Author Rebuttal · Authors · 2026-03-31
>
> **Comment 1:** \
> A quantification of the inference cost of multi-round, multi-role debate for lightweight LLMs with limited context windows, along with an analysis of latency and throughput.
>
> **Response:** \
> Thank you for the comment. We note that we do quantify inference cost in the paper, although we agree this could have been discussed more clearly. First, our evaluation already includes a broad range of model sizes and context capacities, from lightweight models such as LLaMA-3.2-3B-Instruct, Mistral-7B-v0.2, and Phi-3.5-mini, to larger open models (Gemma-3-12B, Qwen3-8B, DeepSeek-R1-Distill-LLaMA-8B, GPT-OSS-20B) and frontier systems (Grok-4 R1-685B, Gemini-2.5-Pro). Across this spectrum, we observe that debate consistently improves performance, including for smaller models with more limited capacity and context windows.
>
> Second, regarding inference overhead, the paper provides token-based cost analysis in the main text and inference-time latency and cost estimates in Appendix C.12, including the hardware used for experiments. These analyses were intended to make the added cost of multi-round debate explicit. We will revise the rebuttal and camera-ready version to more directly reference these results and to clarify the practical trade-off between performance gains and additional inference cost.
>
>  ---
>
> **Comment 2:** \
> Analysis of long-term memory generalization and out-of-distribution robustness, including potential retrieval failures in TLTM and catastrophic forgetting in CLTM.
>
> **Response:** \
> We thank the reviewer for the comment. We include an out-of-distribution generalization study in Appendix C.3, where we test whether safety insights learned from HarmBench and CoSafe transfer to two unseen and more challenging jailbreak benchmarks: Aegis-2 (~1k test instances) and WildJailbreak. As shown in Table 5, the learned insights improve Qwen’s safety performance on both datasets, indicating that the memory is not simply overfitting to the original benchmarks and does generalize to new prompts.
>
> At the same time, we agree that the remaining non-zero error rates show that the system is still vulnerable, and that memory retrieval and generalization are not perfect. We will make this limitation more explicit in the revision. We also agree that a more targeted analysis of TLTM retrieval failures and CLTM forgetting would further strengthen the paper. Our current memory is learned from a relatively modest set of examples (approximately 1.4k from HarmBench and 400 from CoSafe), so a larger-scale study with longer debate histories and explicit stress tests for failure modes is a natural next step. We will clarify that the current results provide initial evidence of out-of-distribution robustness, while a full characterization of long-term memory failure cases remains future work.
>
> ---
>
> **Comment 3:** \
> Whether the proposed framework can be extended beyond the current setting to multimodal or tool-using agents.
>
> **Response:** \
> Thank you for this thoughtful suggestion. Yes—the core idea of our framework is more general than the specific safety setting studied here. The multi-agent debate process is designed to enable agents to propose, critique, recognize failures, and revise, and this mechanism can naturally extend to other agentic settings.
>
> For **tool-using agents**, a direct extension would be to have agents debate candidate tool calls, action sequences, or intermediate plans before execution, with the goal of improving correctness and reliability rather than safety alone. The main additional requirement is an evaluator that can reliably identify tool-use failures (e.g., incorrect tool selection, invalid arguments, or unsuccessful execution outcomes). While this is less straightforward than flagging unsafe responses, it is feasible using execution traces, environment feedback, or task-specific validators.
>
> For **multimodal agents**, the same principle applies as long as both the debating agents and the evaluator can access the relevant modalities or modality-specific feedback signals.
>
> We agree this is a promising direction for future work and will clarify in the revision that the framework is not inherently limited to safety-focused text interactions.

---

> > ### Author Rebuttal · Reviewer_f48w · 2026-04-04
> >
> > The rebuttal from the authors have resolved my concerns (though some questions may need further justifications). I tend to keep my rating.

---

> > > ### Author Response · Authors · 2026-04-06
> > >
> > > Thank you very much for recognizing that your concerns have been adequately addressed. **We have provided additional justification, including a new ablation analysis**, to further clarify the following minor question that may still remain.
> > >
> > > ---
> > > **Followup Comment:**\
> > > Further analysis of long-term memory reliability, including possible retrieval failures in TLTM and the risk of catastrophic forgetting in CLTM.
> > >
> > >
> > > **Response:**\
> > > For **TLTM**, we further evaluate retrieval reliability explicitly. TLTM stores feedback generated in prior debate rounds when the agent makes a mistake. To evaluate whether this memory can be effectively reused, we test whether the corresponding past feedback is retrieved when the same or a semantically similar question is queried again.
> > >
> > > Concretely, we evaluate on the HarmBench dataset and its corresponding TLTM. For each stored <question, feedback> pair, we embed the question, retrieve the top-K nearest neighbors from TLTM and check whether the associated feedback appears in the returned set. We report standard retrieval metrics—Hit Rate, MRR, and NDCG—along with the cosine similarity statistics to assess how close the retrieved memory is to the expected one:
> > >
> > > | Top-K | Hit Rate | MRR | NDCG | Expected Sim | Best Sim | Sim Gap |
> > > |------:|---------:|----:|-----:|-------------:|---------:|--------:|
> > > | 5     | 0.8125   | 0.6797 | 0.7128 | 0.4051 | 0.4254 | 0.0203 |
> > > | 10    | 0.8900   | 0.6900 | 0.7390 | 0.4051 | 0.4254 | 0.0203 |
> > >
> > > These results show that TLTM retrieval is in general reliable. Increasing K from 5 to 10 improves the hit rate from 81.25% to 89.0%, while MRR and NDCG remain strong, suggesting that **relevant memories are usually retrieved and ranked well**.
> > >
> > > Importantly, the similarity gap between the expected feedback and the top retrieved result is very small (~0.02). This shows that even when the exact feedback is not the top-1 match, the retrieved memory is typically highly similar. This is especially relevant in HarmBench, where many prompts share closely related safety failure modes. As a result, retrieval errors are often not catastrophic: if the exact past feedback is missed, the agent still tends to receive a highly relevant corrective signal.
> > >
> > > For **CLTM**, In the experiments reported in the paper, CLTM is updated after every 10 newly collected feedback instances. Importantly, each update fine-tunes the LoRA adapter on **all feedback accumulated up to that point**, rather than only on the most recent batch. That is, the training set is cumulative, not streaming-only. This design helps mitigate catastrophic forgetting because **earlier feedback is revisited at every update, reducing the likelihood that newly incorporated corrections overwrite previously learned behaviors**. We will make this training protocol clearer in the revision, as it is an important detail for understanding the stability of adaptation in CLTM. Thank you very much for your constructive comment.
> > >
> > > We will further clarify two points in the revision: (1) TLTM retrieval failures are relatively infrequent and often softened by semantic overlap among feedback items, and (2) CLTM is updated cumulatively over the full feedback set collected so far, which is intended to prevent catastrophic forgetting of earlier feedback.
> > >
> > > ---
> > >
> > > Thank you very much for your thoughtful comments. We believe that addressing them has strengthened the paper. If you feel that your main comments have been satisfactorily addressed, we would greatly appreciate it if you could consider reflecting this in your overall score. Thank you again for your time and consideration.

---

### Official Review · Reviewer_jqQH · 2026-03-08

**Soundness:** 2
**Presentation:** 3
**Significance:** 3
**Originality:** 3
**Overall Recommendation:** 4
**Confidence:** 4

**Summary:**

This paper proposes RedDebate, a novel multi-agent debate framework that provides the foundation for Large Language Models (LLMs) to identify and mitigate their unsafe behaviours.By combining collaborative argumentation with persistent memory, the system aims to uncover latent vulnerabilities through automated red-teaming and prevent future unsafe outputs. In paper's methodology, authors explore three primary debate structures:
- Peer Refinement Debate (PReD): multiple peer agents with identical roles respond in parallel to a red-teaming prompt
- Devil–Angel Refinement Debate (DAReD): adds two auxiliary agents, named Devil agent and Angel agent. Devil agent generates a rejection which critically challenges the response, acting as a skeptical adversary independent of the actual safety status. On the other hand, Angel agent provides supportive reinforcement, which explicitly encouraged the original response.
- Socratic Refinement Debate (SReD): agent assumes a distinct questioning role to critically examine the responses provided in previous rounds. It highlights reasoning gaps, uncovers implicit assumptions, and demands evidence or clarification to promote deeper reflection.

For memory, authors used four variations of  Long-term memory (LTM):
- Textual Long-Term Memory (TLTM): generated natural language feedback is incorporated into the agent’s prompt to make the agent aware of previously learned lessons.
- Continuous Long-Term Memory (CLTM): stores feedback directly within the LLM’s parameters.
- Unified Long-Term Memory (TLTM+CLTM): uses strengths of TLTM and CLTM , with CLTM reinforcing textual memory, much like how working memory in humans can enrich reading comprehension when presented with explicit textual knowledge and thereby facilitate effective decision-making.
- Guardrails Long-Term Memory (GLTM): automatically encode agents’ past unsafe experiences as guardrails.

In experimental setup, Harmbench and CoSafe are used as datasets. Debate effectiveness is measured using two metrics:
- Error Rate: measures proportion of safe responses among all responses by that agent across all prompts and rounds.
- Agreement Rate: quantifies the proportion of transitions where a response changes from unsafe to safe across consecutive rounds.

Debates were conducted over three rounds. Two triads of debator agents used namely: (Mistral-7B-v0.2(Mis.), LLaMA-3.2-3B-Instruct (Lla.),Phi-3.5-mini (Phi)) and (Gemma-3-12B
(Gem.), Qwen3-8B (Qwe.), Deepseek-R1-Distill-LLaMA-8B (R1)).GPT-4o-mini was used as feedback generator and socratic agent,and all safety evaluations performed using LlamaGuard.
For TLTM, feedback is embedded with OpenAI’s text-embedding-3-large, and the top five entries are retrieved at inference via cosine similarity. The CLTM employs LoRA-based adaptation on the debater’s attention layers. For GLTM, we generate Colang guardrails by prompting GPT-4o.
Scores reported on just debate alone shows SRedD produces least error rate of 21% as compared to 38.7% on Standard Prompting for HarmBench dataset. Subsequently, in the scores reported for Debate with memory integration shows GLTM based approach produces the least error rate of 0.2% as compared to 28.9% on SReD for HarmBench dataset. Paper reports that additional compute for self-critique does not match debate’s safety gains. The ablation studies also highlight that instruction-tuned models are required for meaningful participation; base models fail to engage, while overly aligned models simply refuse unsafe prompts without disrupting the broader debate. Finally, the authors determine that increasing the number of agents boosts diversity, with three sequential debate rounds serving as the optimal setup for minimizing errors.

**Compliance With Llm Reviewing Policy:**

Affirmed.

**Final Justification:**

I would like to thank the authors for their continued engagement and for taking my remaining critiques seriously during this discussion phase. The effort to run a new ablation study on such short notice is highly commendable.

After reviewing this final set of responses, I am finalizing my score as a Weak Accept (4).
The rebuttal successfully addresses several of my concerns, but a few structural limitations remain that prevent a higher score:
- Disabling the "thinking mode" for DeepSeek-R1 and Qwen is the main reason I am not raising my score to a full Accept. While I completely understand author's engineering rationale (the 512-token limit), having a valid excuse for a limitation does not remove the limitation. By turning off the CoT reasoning, the framework bypasses the most critical safety challenge of modern frontier models: auditing hidden reasoning tokens. Consequently, the paper proves this framework works for standard instruction models, but fails to prove it can handle true reasoning models, which significantly limits the framework's scope and impact.
- I appreciate author's honesty in acknowledging the Data Poisoning vulnerability. Moving this to the Limitations section is the correct academic response. However, deploying an online memory system (CLTM/TLTM) with an unmitigated vulnerability to adversarial poisoning means the system is not yet robust enough for safe, real-world deployment.

**Key Questions For Authors:**

Questions and clarifications from authors are embedded along with Weaknesses i had listed earlier.

**Limitations:**

For Limitations, no, limitations mentioned by author are not adequate as i have highlighted as part of the weakness section, like for example: brittleness of hard-coded guardrails and subsequent over-refusal is not discussed. Also possibility of data poisoning with memory integration is not discussed as part of limitation.
For negative societal impact, yes, authors discussed in the paper.

**Strengths And Weaknesses:**

## Strengths:
- Paper's proposed design of combining memory with AI debates to keep models safe over time is smart  and efficient, as it helps the model handle brand-new attacks (zero-day jailbreaks) quickly, reducing the high cost and time of retraining the whole system from scratch.
- Good choice of Constitutional AI (CAI) based self-critique and Best-of-N (BoN) sampling baselines, which are highly relevant and stringent benchmarks for reasoning and safety tasks.
- Variation of Long term memory(LTM) listed in paper such as Textual(TLTM), Parametric(CLTM), Unified(TLTM+CLTM) and Guardrail(GLTM) memory, provides a robust memory taxonomy which can be reusable across AI Safety and alignment research.
- Implementations of LTMs and agent prompts are shared by authors in the paper to help in reproducibility of the paper.
- tracking of Agreement Rate (AGR) by authors is a thoughtful way to measure true debate diversity and prevent agents from just echoing each other.
- Choice of heterogeneous triad of models is a good design choice made by authors is the ideal experimental choice to prevent identical models collapsing into agreeing with one another.

## Weakness:

- Multi-agent debate requires more compute and tokens , as compared to a single agent. In order to justify the high token usage, authors tried to justify it by comparing it to self-critique with figure 5 which has major issues like more points on plot for self-critique as compared to debate, and whenever they're inline with same budget, correspondingly error rate seems pretty low or insignificant.
- Comparing SReD to a Best-of-N (BoN) baseline of N=3 is an unfair test. SReD uses vastly more computing power and time because it runs multiple models over several rounds, uses an external guide, and relies on reasoning models (like Qwen and DeepSeek-R1) that generate massive amounts of internal "thinking" tokens. Because SReD uses orders of magnitude more resources than three simple parallel samples, it is impossible to tell if the architecture itself is actually better. The authors must scale N in the BoN baseline to match SReD's total compute or token budget for a fair comparison.
- As listed in Subsection 4.3 ,implementation uses GPT-4o-mini as Socratic agent, which may be comparable in parameter count (~8B) to the triad models, but do possess significantly higher reasoning and proprietary safety-alignment capabilities. In the process of gpt-4o-mini guiding much smaller and less capable debater models (3B-12B), it makes it hard to tell if the safety improvements come from the multi-agent debate itself, or if the gpt-4o-mini being the stronger model is just doing the heavy lifting by coaching the weaker ones. To prove the debate framework actually works on its own, the authors should run a test (ablation) where a smaller and less capable model acting as the Socratic guide.
- While the SReD+GLTM method gets the best safety scores, it uses strict, hard-coded rules to block inputs. Rigid filters like this can cause a model to do over-refusal or to become overly cautious and reject perfectly safe, normal requests. The authors need to test the framework on a dataset of harmless prompts to check for false positives. A low safety error rate isn't a practical success if the model is just refusing to answer regular questions.
- One of the triad of debater agents consists of two thinking models, namely Qwen3-8B (Qwe.), Deepseek-R1-Distill-LLaMA-8B (R1), whic does perform uses internal "chain-of-thought" for self-correction.
- The paper does not explain how the internal reasoning of Qwen and DeepSeek-R1(debator agents in one of triad) is handled. It fails to mention whether the Socratic agent sees the models' internal <think> tags. If this reasoning is kept hidden, GPT-4o-mini is just critiquing the final answer without seeing the logic behind it, which invalidates the Socratic approach. On the other hand, if the reasoning is shared, the context window would suffer from extreme token bloat. The authors need to be transparent about how this chain-of-thought routing actually works in their setup..
-  If the framework updates its Textual (TLTM) and Parametric (CLTM) memory based on ongoing adversarial interactions, it is highly vulnerable to Data Poisoning. A sophisticated attacker could possibly engage the model in subtle, multi-turn debates designed to slowly inject flawed "safety lessons" into the memory database, eventually tricking the system into permitting a jailbreak.

---

> ### Author Rebuttal · Authors · 2026-03-31
>
> **Comment 1:** \
> Extending the BoN baseline with compute matching to SReD.
>
> **Response:** \
> Thank you for your considerate suggestion. We add BoN (N=5) so agents average token usage exceeds the SReD budget. The resulting token usage is:
> |Method|Gemma|Qwen|R1|
> |---|---:|---:|---:|
> |SReD|1977|1868|1545|
> |Bo5|2538|2430|2036|
>
> Even with higher budget, SReD outperforms Bo5:
> - Best: 29.7/28.8/29.3/**31.0**
> - Avg: 42.7/43.8/36.9/47.3
> - Worst: 53.6/54.3/43.1/63.5
> - SReD: **28.9/24.8/28.6**/33.4
>
> We also like to clarify that BoN “best try” is an optimistic oracle rather than a realistic deployment baseline, as it marks an example safe if any of the five samples is safe. While useful as an upper bound, it does not reflect the reliability of a system; the average-case BoN is more representative for practical comparison. When scaling Bo3 to Bo5, the average-case error changes minimally: Gemma (-0.58%), Qwen (+0.18%), and R1 (+0.03%). This suggests that adding more independent samples does not meaningfully improve safety, supporting our claim that gains come from structured deliberation rather than increased token usage.
>
> ---
>
> **Comment 2:** \
> The reviewer asks whether Fig. 5 fairly compares to self-critique (SC), noting that SC includes more plotted points and that gains at similar token budgets may appear small.
>
> **Response:** \
> The difference in the number of plotted points arises from how the methods consume tokens. Since SC iterations are relatively shorter, we allow it to take more revision steps until its cumulative token usage approximately matches that of debate. This is why SC agents have more points in the plot. Viewed from this perspective, Fig. 5 reveals a consistent pattern:
>
> **SC often plateaus or becomes unstable, while debate is more reliable.** \
> For some models (notably Llama and Phi, consistent with Fig. 3), SC can even increase error or unsafety at intermediate steps, whereas debate shows more consistent error reduction. More broadly, self-critique tends to saturate even when given additional revision opportunities under the same overall token budget. For example, in Qwen, SC requires many more revisions (up to 9) to approach the performance of a 3-round debate. In R1, no further corrections occur after turn 3, and performance even diverges in the final turn.
>
> Therefore, our intended claim is not that debate is cheaper per attempt—it is not—but that its additional token cost is offset by better and more stable safety correction at matched budgets.
>
> ---
>
> **Comment 3:** \
> Whether a strong auxiliary model is the main cause of the gains of the debate framework.
>
> **Response:** \
> First, our study does not rely solely on the Socratic setting to support the main claim. We also include PReD and DAReD, which do not use GPT-4o-mini or any auxiliary Socratic agent. As shown in Tables 1 and 10, these non-Socratic debate settings already improve over the corresponding no-debate baselines. This demonstrates that the core benefit does not depend on GPT-4o-mini: even without an external guide, agents can revise unsafe responses.
>
> Second, in Socratic agent is not used to directly provide safety corrections or “coach” the models toward safer answers. Its role is deliberately constrained to asking questions, challenging assumptions, and eliciting justification. This is reflected in the Socratic prompt (Fig. 21), where the agent is instructed to probe logic rather than prescribe safe responses. The actual safety feedback is generated after the debate by a separate feedback generator. For this reason, we intentionally use a stronger model for the Socratic role to reliably surface inconsistencies and encourage self-correction.
>
> ---
>
> **Comment 4:** \
> Details on how internal CoT from Qwen and R1 is handled.
>
> **Response:** \
> In our experiments, we disable thinking mode. Thus, in the reported setup, no CoT is generated or shared between agents, and there is no hidden-reasoning routing issue or context inflation from passing internal reasoning traces. We choose this design intentionally to maintain a consistent debate interface across all triads.
>
> ---
>
> **Comment 5:** \
> Whether the improved safety of GLTM comes at the cost of over-refusal.
>
> **Response:** \
> We would like to clarify that we evaluate this issue in Appendix C.7 (“Helpfulness After Memory Integration”). We also extend the analysis using XSTest on more challenging prompts, as discussed in Comment 1 for Reviewer FaZL.
>
> ---
>
> **Comment 6:** \
> Robustness of the memory component against data poisoning attacks.
>
> **Response:** \
> Memory poisoning is an important concern. In our framework, memory updates are mediated by evaluator and feedback agents that independently assess the **full debate history**. This means a successful attack must evade both these agents and the debaters, making subtle multi-turn poisoning significantly harder. While not eliminating the risk, as models are not guaranteed to be perfectly robust, this design substantially reduces its likelihood.

---

> > ### Author Rebuttal · Reviewer_jqQH · 2026-04-01
> >
> > I would like to thank the authors for their rebuttal. I will be raising my overall recommendation score from 3 to 4.
> > While most of the points are addressed but I still see some limitations not addressed fully, thus supporting my reasoning for raising the score to 4 but not 5 :
> >  - with additional experiments BoN experiments conducted with N=5, it does demonstrate that SReD outperforms a compute-matched BoN baseline on the average-case error rate.
> > - Clarification provided for token consumption rate and the number of plotted points for SC does make sense and help clarify the confusion around SC having more points on plot.
> > -  Authors mention that they disabled thinking mode in Qwen and R1 to make a consistent debate interface. However inclusion of models like Deepseek-R1 and Qwen does set the expectation of evaluating their thinking abilities via which they achieved higher State-of-the-art scores on various benchmarks. But disabling the thinking mode also bypass one of the critical safety challenges with these models, which is auditing their internal CoT. Therefore disabling the thinking mode remains a significant limitation as this framework evaluates instruction tuned models but not the thinking models (with the thinking disabled).
> > - PReD and DAReD configurations does make safety improvements without relying on GPT-4o-mini, but highest-performing configuration (SReD + GLTM) relies on gpt-4o-mini model for both Socratic guidance and Colang guardrail generation. So this dependency on a more aligned model still remains a limitation for  framework's autonomy.
> > - Authors do mention that multi-agent mediation acts as a defense to data poisoning, however there is no empirical evaluation done to substantiate that claim and does make CLTM/TLTM modules vulnerable in real world deployments.

---

> > > ### Author Response · Authors · 2026-04-05
> > >
> > > We truly appreciate the reviewer’s additional thoughtful comments. Below, we provide further responses to and experiments for the comments. We sincerely hope they could be further considered if they have addressed the questions. Thank you very much.
> > >
> > > ---
> > >
> > > **Comment 1**
> > > Reliance on GPT-4o-mini for Socratic guidance and guardrail generation.
> > >
> > > **Response:**
> > > We agree that the best-performing configuration (SReD+GLTM) still uses GPT-4o-mini. Our additional ablation shows that the framework’s safety gains are **not solely dependent** on it.  To show that, we replaced GPT-4o-mini with **LLaMA3-3B-Instruct** as the Socratic guide and measured the stepwise error rate across debate rounds. We can still observe consistent safety improvement over the course of deliberation:
> > >
> > > - **Gemma:** 34.2→12.9→9.4
> > > - **Qwen:** 41.6→22.8→14.9
> > > - **R1:** 66.8→56.9→57.9
> > >
> > > The results demonstrate that with a weaker open model as the Socratic guide, agents can self-correct in debate by using probing questions to refine their own reasoning.
> > >
> > > Our detailed transcript analysis further suggests a potential downside of using a weaker guide, although the issue concerns logical consistency and context loss rather than safety directly. LLaMA3-3B often asks useful follow-up questions and challenges assumptions in early rounds, but it often **loses debate context in later rounds** and produces generic or irrelevant replies (e.g., refusals). In contrast, GPT-4o-mini is more consistent at maintaining context and sustaining the probing process, making it a stronger Socratic facilitator.
> > >
> > > In addition, for **GLTM**, as explained in Ln 245, GPT is **not used to make safety decisions or directly enforce safer behavior**. It serves as a **translator** that converts already-generated textual feedback into executable **Colang** rules with a correct syntax. The actual safety-relevant behavior—intent matching, refusal, redirection, and response constraints—is handled by **NeMo Guardrails** and inferring the debater agent. We will make this clearer and point readers to examples in Fig. 23, illustrating that this step is primarily syntactic compilation rather than safety augmentation.
> > >
> > > In short:
> > > 1. **Socratic safety gains persist with a weaker open model**, albeit with some potential for context loss; and
> > > 2. **GPT-4o-mini’s role in GLTM is primarily the syntactic translation of feedback into Colang, not direct safety supervision**.
> > >
> > > ---
> > >
> > > **Comment 2**
> > > Implications of disabling thinking mode for Qwen and DeepSeek-R1.
> > >
> > > **Response:**
> > > Our objective of including Qwen and R1 was not so much for evaluating or maximizing their peak reasoning-mode performance, but testing if multi-agent debate improves safety by bringing together models with **different perspectives, safety tendencies, and interaction styles.** This heterogeneity is useful for our framework: in the experiments Qwen was usually more verbose and debate-oriented, while R1 was relatively more permissive on unsafe prompts, creating opportunities for cross-agent correction.
> > >
> > > We also capped each model’s response at **512 new tokens** so that no debater could consume a disproportionate share of the budget. With reasoning mode enabled, we observed some models use much or all of this limit on intermediate reasoning, leaving little room for the actual debate response or final answer. Disabling reasoning mode hence gave a cleaner, more comparable setup and ensured the token budget was spent on the debate itself.
> > >
> > > We will clarify in the paper that our results apply to chat/instruction-mode behavior, and evaluating debate with reasoning mode enabled is an important direction for future work. Thank you!
> > >
> > > ---
> > >
> > > **Comment 3**
> > > The rebuttal notes that multi-agent mediation may help reduce memory poisoning, though it is not empirically evaluated.
> > >
> > > **Response:**
> > > Memory poisoning is an important consideration for a system with online memory updates. We did not include an empirical evaluation of this issue in the submission, as it was not part of our original experimental scope. At the current point, we do not add a full poisoning study since it would likely entail the development of a novel data poisoning approach working with multi-agent, multi-memory setups that may warrant a new paper (many popular memory poisoning methods like [1] target single agent single memory systems). For this reason, we intended this point as a potential advantage of the mediated update design, rather than as an empirically established robustness claim.
> > >
> > > We thank the reviewer. We will clarify this explicitly, avoid stating the protection offered by mediation alone, and note that poisoning-resilient memory update mechanisms and adversarial evaluation are important directions of future work.
> > >
> > > If you think your main comments have been addressed, we humbly ask if you could further consider reflecting that in your score.
> > >
> > > [1] Dong et al. Memory Injection Attacks on LLM Agents via Query-Only Interaction. NeurIPS '25

---

### Official Review · Reviewer_FaZL · 2026-03-12

**Soundness:** 3
**Presentation:** 3
**Significance:** 2
**Originality:** 3
**Overall Recommendation:** 4
**Confidence:** 4

**Summary:**

RedDebate introduces an automated safety framework that leverages collaborative argumentation among multiple language model agents to identify and mitigate unsafe reasoning patterns at inference time. The system incorporates diverse long-term memory modules—including textual, parametric, and programmatic guardrails—to store distilled safety lessons and prevent the recurrence of identified failure modes. Empirical evaluations across various safety benchmarks demonstrate that this multi-agent interaction, combined with persistent memory, significantly enhances model alignment and response safety compared to isolated self-revision or standard prompting.

**Compliance With Llm Reviewing Policy:**

Affirmed.

**Final Justification:**

Most of my concerns have been addressed. I have increased my initial score to weak accept.

**Key Questions For Authors:**

See Weaknesses.

**Limitations:**

Yes

**Strengths And Weaknesses:**

# Strengths

- The paper presents an interesting observation that structured collaborative argumentation among multiple agents can effectively surface and correct safety vulnerabilities that single-agent reflection might overlook.
- The proposal to utilize diverse long-term memory types—including textual, parametric, and programmatic guardrails—to preserve and reuse safety-relevant insights from past debates is a novel and intriguing approach for continuous safety alignment.
- The research is well-motivated by the limitations of current training-time alignment methods and is presented with a clear, logical structure, supported by detailed framework overviews.
- The experiments cover an extensive range of models and diverse safety datasets, effectively demonstrating the performance gains of the proposed RedDebate framework across various scenarios.

# Weaknesses

- The framework requires multiple agents to generate responses across several rounds for every user query. Although the authors argue that this form of “inference-time scaling” is justified, the cumulative latency and computational overhead—especially in multi-turn attack scenarios—remain significantly higher than those of single-agent guardrails or standard refusal mechanisms.

- The advantages over single-agent baselines are not sufficiently established to justify the additional cost. The authors only consider Self-Critique as a single-agent baseline; however, many alternative approaches exist, such as lightweight guardrail models. Furthermore, several recent works have explored diagnostic approaches in which the harmfulness or intent of a user query is analyzed before generating a response [1,2,3,4,5]. These approaches allow a single LLM to detect harmful queries without requiring an expensive debate procedure.

- Although HarmBench and CoSafe are used for evaluation, the analysis lacks depth with respect to state-of-the-art jailbreak techniques and multi-turn adversarial attacks. While some jailbreak experiments are mentioned in the appendix, they are not central to the main safety claims.

- Although the TriviaQA results show maintained accuracy, this benchmark alone is insufficient to rigorously evaluate both over-refusal and general capability. For assessing over-refusal, the authors should consider additional datasets specifically designed for that purpose, such as [6]. To evaluate whether general capabilities are preserved, the experiments should include a broader range of tasks beyond question answering, such as summarization and reasoning.

- The current framework primarily focuses on single-prompt debates. Its effectiveness and cost efficiency under multi-turn dialogue attacks remain unclear and may be difficult to maintain in practice.

[1] Intention analysis makes llms a good jailbreak defender
[2] Deliberative alignment: Reasoning enables safer language models
[3] Star-1: Safer alignment of reasoning llms with 1k data
[4] Reasoning-to-defend: Safety-aware reasoning can defend large language models from jailbreaking
[5] R1-ACT: Efficient Reasoning Model Safety Alignment by Activating Safety Knowledge
[6] XSTest: A Test Suite for Identifying Exaggerated Safety Behaviours in Large Language Models

---

> ### Author Rebuttal · Authors · 2026-03-31
>
> **Comment 1:** \
> Additional over-refusal benchmark.
>
> **Response:** \
> Thank you for your constructive suggestion. We have added the experiments on XSTest below.
>
> |Model|Memory|Compliance|Δ Comp|Full Refusal|Partial Refusal|
> |---|---|---|---|---|---|
> |**Mistral**|—|73.6|—|16.0|10.4|
> ||TLTM|72.8|-0.8|15.2|12.0|
> ||CLTM|74.0|+0.4|19.2|6.8|
> ||CLTM+TLTM|59.6|-14.0|30.4|10.0|
> ||GLTM|80.6|+7.0|12.3|7.1|
> |**Llama**|—|82.4|—|14.0|3.6|
> ||TLTM|83.2|+0.8|13.6|3.2|
> ||CLTM|91.6|+9.2|7.6|0.8|
> ||CLTM+TLTM|85.6|+3.2|12.8|1.6|
> ||GLTM|91.2|+8.8|8.4|0.4|
> |**Phi**|—|80.8|—|15.2|4.0|
> ||TLTM|80.8|0.0|14.8|4.4|
> ||CLTM|80.8|0.0|18.0|1.2|
> ||CLTM+TLTM|72.8|-8.0|26.0|1.2|
> ||GLTM|80.5|-0.3|16.5|3.0|
>
> The results show that adding LTM does not lead to a meaningful reduction in compliance behaviour relative to the base models. Across all three model families, the observed changes are not consistently in the direction of increased refusal, except for Hybrid memory in Mistral and Phi. While it works well for Llama, we interpret this as being associated with a potential limitation of this memory type, which may be prone to over-regularization if not carefully tuned.
>
> ---
>
> **Comment 2:** \
> Whether the extra compute of multi-agent debate is justified relative to refusal guardrails and query-diagnosis approaches.
>
> **Response:** \
> We respectfully argue that the claim of our paper is not that debate should replace lightweight guardrails or refusal-based methods in all settings. Rather, the paper studies a different capability: **whether models can recognize and correct their own safety blind spots through interaction**, instead of only detecting a harmful query and refusing it. This distinction motivates our single-agent correction baselines. We compare against Self-Revision, Self-Critique, and parallel Best-of-N, all of which give the model multiple opportunities to reconsider its answer. With a comparable token budget, interactive deliberation elicits corrections that single-agent reasoning often misses. This is reflected in our agreement-rate analysis (Table 1) and the step-wise correction trends (Figures 4 and 5).
>
> We also agree that lightweight safety mechanisms are practically important. For this reason, our framework is not intended as a costly, always-on alternative to guardrails. A novel mechanism we introduce is the Guardrail Long-Term Memory (GLTM), which converts insights discovered during debate into lightweight, updatable guardrails that can be efficiently reused at deployment time. Importantly, this is **complementary to refusal and intent-detection methods**, not in conflict with them. In fact, our system already incorporates such a mechanism via NeMo Guardrails. As explained in line 245, NeMo Guardrails works by analyzing the intent of the user query and matching it against available guardrail rules, which then trigger refusal, redirection, or response constraints. In our framework, debate is used to discover and refine missing safety knowledge, and GLTM stores that knowledge so it can later be applied efficiently through NeMo’s intent-based guardrailing. In other words, the pipeline is: learn from failures via debate, then deploy the learned safety rules through a lightweight guardrail mechanism. Thus, methods based on refusal or intent analysis are are natural downstream beneficiaries of our framework.
>
> We appreciate the cited references and will revise the paper to position our method more clearly with respect to them.
>
> ---
>
> **Comment 3:** \
> Suggestions on considering broader jailbreak attacks.
>
> **Response:** \
> To clarify the intended scope, one of our objectives is to demonstrate that multi-agent debate improves a model’s ability to self-identify and reason about safety violations, rather than proposing a universally robust defense against all potential jailbreak attacks. To enable controlled and comparable evaluation, we conduct our main experiments on relatively clean and standardized benchmarks. This choice allows for consistent comparisons with prior work and isolates the effect of the debate mechanism.
>
> Jailbroken data inherently introduces additional complexity, as different attack forms can affect models in unpredictable ways depending on how they are optimized. Given the combinatorial design space in our setup—multiple model triads, memory variants, and debate strategies—incorporating a wide range of jailbreak attacks would lead to an exponentially larger evaluation space, making controlled analysis difficult within the scope of this paper. That said, the appendix includes experiments with diverse jailbreak strategies (e.g., WildJailbreak, covering forceful phrasing, deceptive framing, role-play scenarios, and etc.) as supporting evidence.
>
> We view jailbreak defenses as largely orthogonal to our work: most existing approaches focus on blocking specific attack patterns or inducing refusal, whereas our method aims to help models understand why a response may be unsafe and retain that insight for future interactions.

---

> > ### Author Rebuttal · Reviewer_FaZL · 2026-04-03
> >
> > Most of my concerns have been addressed. I increased my initial score.

---

### Decision · Program_Chairs · 2026-04-30

**Decision:**

Accept (regular)

**Comment:**

All four reviewers recommend weak accept and declared concerns fully resolved after rebuttal.
The paper presents a novel unification of multi-agent debate and automated red-teaming with persistent memory, showing consistent safety improvements across diverse models. The rebuttal added a compute-matched BoN baseline confirming SReD's advantage, XSTest results showing memory does not increase over-refusal, and an ablation with a weaker open Socratic guide showing gains persist without GPT-4o-mini.
Remaining limitations, including thinking mode disabled for reasoning models, GPT-4o-mini dependency in the best-performing configuration, and unvalidated poisoning resistance of memory modules, are acknowledged and appropriate for future work. Reviewer consensus is clear.